# Soil Nematode Communities in Managed and Natural Temperate Forest

Andrea Čerevková [1,*], Marek Renčo [1], Dana Miklisová [1] and Erika Gömöryová [2]

1   Institute of Parasitology SAS, Hlinkova 3, 040 01 Košice, Slovakia; renco@saske.sk (M.R.); miklis@saske.sk (D.M.)
2   Faculty of Forestry, Technical University in Zvolen, TG Masaryka 24, 960 01 Zvolen, Slovakia; erika.gomoryova@tuzvo.sk
*   Correspondence: cerev@saske.sk

**Abstract:** Forest management and the stand age play key roles in determining the composition of soil biota, including nematodes. We analysed the effect of the interaction between stands of natural forest and stands influenced by human activity on nematode communities, necessary for realistically assessing the specific potentials of forest soils, plant protection, forest management, and land use management. Nematode communities were evaluated and compared in managed beech and spruce forests in three age classes (0–20, 40–60, and 100–120 years old) and an unmanaged old-growth temperate forest. A total of 51 nematode genera were found in the forests. The number of nematode genera was the highest (46) in European beech forests, dominated by *Rhabditis* and *Filenchus*. In contrast, the number of nematode genera was the lowest (37) in a Norway spruce forest, but where nematode abundance was the highest due mostly to the high abundance of bacterivorous nematodes such as *Acrobeloides*, *Plectus*, and *Rhabditis*. The unmanaged old-growth forest had the lowest nematode abundance and total biomass but the highest abundance of herbivorous nematodes of the order Tylenchida, especially *Filenchus*, *Malenchus*, and *Paratylenchus*, and a high abundance of identified genera of predators. The number of identified nematode genera, abundance, total biomass, and diversity index were the highest in young 0–20-year-old stands, and the lowest in 100–120-year-old stands. Enrichment, structure, and basal indices were influenced by both the stands and the ages of the forests.

**Keywords:** European beech; Norway spruce; managed forests; old-growth mixed forest; age class; Nematoda

## 1. Introduction

Soils and plants in forest ecosystems are tightly linked, due especially to the long-term influence of forest stands on the soil. Soil conditions determine plant growth to some extent, which is why plants are widely used as indicators or for estimating soil and site properties [1,2]. Plants, though, strongly affect soil properties, including the characteristics of communities of soil biota. Trees affect soil via both above- and belowground resource-based mechanisms (e.g., litter input on the soil surface, belowground deposition of root exudates, dead roots), and by altering abiotic conditions (e.g., by leaf fall, shading, frost protection, transpiration) [3]. Trees can strongly affect the physical structure of the soil, water flow, soil pH, and the contents of soil organic matter and nutrients [4,5]. Changes in soil properties can consequently affect the quality of living conditions for soil biota, leading to changes in their abundance, biomass, activity, and community structure [6–8].

European beech (*Fagus sylvatica* L.) and Norway spruce (*Picea abies* (L.)) are dominant species in forests in Central Europe, covering 30% of the forested area [9–11]. European beech forests have been exploited more intensively than Norway spruce forests and have been mostly converted to age class systems. Pure stands of Norway spruce have often been cultivated outside its natural range at the expense of European beech throughout the last 200 years [12]. Norway spruce outperforms most other species when water is

abundant, and bark beetles, windthrow, and ice breakage are rare [11,13]. Natural unmanaged forests represent unevenly aged forests characterised by a fine-grained mosaic of different developmental stages of trees due to frequent small-scale disturbances and the presence of large amounts of dead wood and decaying trees [14,15]. After harvesting, mixed mountain forests were often left in a more natural state due to the difficulty of access. High heterogeneity within unevenly aged stands increases biodiversity and stand stability, and such forests can more successfully fulfil their functions than forests with more traditional evenly age stands favoured by policy makers [16]. Studies in the last decade, however, have confirmed that biodiversity is not necessarily higher in forests without human influence [17,18]. This inconsistency has led to ongoing discussions of biodiversity in managed and unmanaged forests [19]. Studies have especially been concerned with plants, birds, or beetles, but only a few studies have investigated soil nematode communities.

Nematodes in forest soil are ubiquitous, abundant, functionally diverse, and very sensitive to environmental changes. They participate in all major trophic levels of the soil food web and are responsible for several processes vital for the correct functioning of soil ecosystems [20,21]. The analysis of the composition of nematode fauna serves as a basis for the ecological assessment of soil [22]. Nematode communities are made up of diverse species that, according to their feeding habits, can be classified into five major groups: plant parasites, bacterial and fungal feeders, predators, and omnivores [23]. The most abundant nematode taxa in forest soil feed on bacteria and, together with fungivorous nematodes, play a key role in the decomposition of organic matter and the cycling of nutrients in the soil [23,24]. Herbivorous nematodes feed on and damage plant roots and can have a negative impact on plant growth. An increase in their abundance or a strong dominance of herbivore species can occur in biotopes with characteristics similar to those in long-term monocultures and is usually associated with soil degradation [23,24]. Omnivorous and predatory nematodes represent the highest trophic level amongst soil microfauna. An increase in the abundance of these groups may be an indication of the naturalness of the environment [23,25]. Analysis of nematode communities provides information on succession, changes in the pathways of decomposition in soil food webs, nutrient status, fertility, soil acidity, and the effects of soil contaminants [26]. Nematodes are useful indicators of soil conditions due to their ecological importance and sensitivity to environmental changes [21,25,26].

In our study, we focused on soil nematode communities in managed (pure beech and pure spruce) stands and an unmanaged old-growth forest, all representing the most typical and broadly distributed forests in mountains of the temperate zone in Europe. The goals of our study were to determine (i) whether and to what extent the management of forest stands affects the abundance and structure of nematode communities compared to unmanaged forests, (ii) whether and how the stands of different age classes alter soil nematode communities, and (iii) whether or not the patterns of nematode responses associated with the stand age and management are the same in spruce and beech ecosystems. We hypothesised that nematode communities would generally be more diverse in the unmanaged forest due to the more heterogeneous stand structure. Environmental conditions and understory vegetation change during the lifecycle of a stand, however; therefore, we expected that the succession of developmental stages would affect nematode communities, and that the pattern of responses would not necessarily be uniform for tree species such as beech and spruce, representing broadleaves and conifers.

## 2. Materials and Methods

### 2.1. Site Description

This study was carried out on Mount Poľana (48°37′ N 19°30′ E) in the Western Carpathian Mountains in central Slovakia (Europe). Mount Poľana is one of the highest European former volcanoes, with an altitude of 1458 m a.s.l., formed mainly of andesite and andesite tuffs rich in soil nutrients. Eutric Cambisol is the most widespread soil type at lower altitudes, with Dystric Cambisols transitioning to Andosols at higher altitudes, where

the presence of allophane has led to a higher capacity of the soil to accumulate organic matter [27]. Soils are deep, with variable contents of rock fragments depending on the parental material. Soils developed from andesite are characterised by a high stone content in the profile, but soils developed from andesite tuffs contain much fewer rock fragments. Mean annual temperature ranges from 2 to 4 °C, and mean annual precipitation ranges between 900 and 1200 mm [28,29], typical for a humid continental climate. The vegetation of the area is very diverse, with >1200 taxa of higher vascular plants. Forests represent the most typical and broadly distributed example of temperate forests in European mountains [29]. The species composition of the native trees has been affected by management practices, but natural forests in a part of the area have been protected as natural reserves since 1972 and as Biosphere Reserves since 1990. Plots were established at sites with similar altitudes of 950 to 1250 m a.s.l. on slopes with southern aspects (ESE to SE).

## 2.2. Sampling Procedure

This study was performed in three types of forest ecosystems: a managed beech forest (BEE), a managed spruce forest (SPR), and an unmanaged mixed forest (UNM).

The managed forests (BEE and SPR) were developed from natural forests, clearcut, and characterised by a single crown layer typical for evenly aged stands. In BEE, European beech (*Fagus sylvatica* L.) from natural regeneration is the dominant tree species, but *Acer* spp. originating from natural regeneration can also occur. In SPR, Norway spruce (*Picea abies* (L.) H. Karst.) trees were planted. The plots in BEE and SPR were stratified by stand age into three age classes: 0–20, 40–60, and 100–120 years of age.

UNM represents an old-growth forest in an untouched area, characterised by diverse spatial structures, heights, and diameters, and composed of mixed tree species dominated by *F. sylvatica*, *Abies alba* Mill., and *Acer pseudoplatanus* L., with occasional *Fraxinus excelsior* L., *P. abies*, and *Ulmus glabra* Huds. [29].

Our study had a total of 45 plots, 15 plots for each forest type (5 plots for each age class in each forest type (BEE and SPR) and 15 plots in UNM). In each plot, five average soil samples, which consisted of four sub-samples from a 1 m² area at a depth of 10–15 cm, were collected in August 2019 using a hand spade. The samples were thoroughly mixed, transferred to the laboratory in plastic bags, and stored at 5 °C until processing.

## 2.3. Analyses of Soil Properties

The soil moisture content of the fresh samples was estimated gravimetrically by oven drying at 105 °C overnight to a constant weight. The chemical soil properties were analysed in air-dried soil samples. Soil pH was measured potentiometrically in a water suspension using a digital pH meter at a 1:2.5 soil/water ratio. The contents of total carbon (C) and nitrogen (N) were determined using a Vario MACRO Elemental Analyzer (CNS Version; Elementar, Hanau, Germany).

## 2.4. Analyses of Nematode Communities

Nematodes were isolated from 100 g of soil using a modified Baermann technique with a set of two cotton-propylene filters for 24 h at room temperature (20 °C), and extracted nematodes were heat killed, fixed in Ditlevsen's solution, and mounted in glycerin [30]. Isolated nematodes were identified at the genus level using a light microscope (Nikon Instruments Europe BV, The Netherlands), original species descriptions, and several taxonomic keys [31–38].

The nematodes were divided into five trophic groups: bacterivores, fungivores, herbivores, omnivores, and predators [25]. The maturity index (MI) for free-living taxa and the plant parasite index (PPI) for plant-parasitic taxa [39] were calculated using the coloniser-persister (cp) value based on life history trails following Bongers and Bongers [40]. The enrichment index (EI), the structure index (SI), the channel index (CI) [41], and the basal index (BI) [42] were calculated. The CI is calculated as the percentage of bacterivores relative to their number plus that of fungivores. The EI, SI, CI, and BI differ in that their calculation involves a weighing system for nematode functional guilds, and in that they

are used to infer the food web complexity and the main pathways of organic matter decomposition [41,42]. The nematodes were assigned to trophic groups [23,25,43]. All indices (MI, PPI, EI, SI, BI, CI) and nematode biomass were calculated using the online programme 'NINJA: An automated calculation system for nematode-based biological monitoring' [43]. The Shannon–Weaver index was used for calculating generic diversity: H'gen = $-\sum(P_i \times \ln P_i)$, where $P_i$ is the proportion of the genus divided by the total nematode abundance in the sample [44]. The dominance of the nematode genera (D, %) was calculated as: $D = (n/s) \times 100$, where $n$ is the total abundance of nematode genera, and $s$ is the total abundance of nematode genera per sample. The frequency of occurrence (F, %) was calculated as: $F = (n_i/s) \times 100$, where $n_i$ is the number of samples containing genus $i$, and $s$ is the total number of samples.

### 2.5. Statistical Analysis

The data were analysed in two ways. All samples were first analysed by forest type (BEE, SPR, and UNM), but samples for only two stands were analysed by both forest type (BEE and SPR) and age class (0–20, 40–60, and 100–120 years of age) due to the unstructured nature of UNM.

Soil moisture content, pH, total C, total N content, and the C/N ratio were analysed untransformed because they satisfied the assumptions of the parametric tests. In contrast, nematode characteristics (total abundance, H'gen, total biomass, and abundance per trophic group) and basic ecological characteristics (MI, PPI, EI, SI, BI, and CI) did not meet the assumptions of the parametric tests; therefore, a Box–Cox transformation was applied using maximum likelihood and a golden search iterative procedure prior to the tests.

One-way ANOVA was used for samples analysed by forest type. Two-way ANOVA with mixed effect and main effect ANOVA (if a mixed effect was not confirmed) was used for samples analysed by both forest type and age class. Each type of forest was analysed separately if a mixed effect of forest type × age was significant (only for H'gen, PPI, and omnivores). Fisher's LSD post hoc test was used to identify differences amongst the age classes. All statistical analyses were performed using Statistica Cz, version 12.0 [45].

## 3. Results
### 3.1. Effect of Forest Management

The soil moisture content was significantly higher in UNM than SPR ($p < 0.05$) and was very similar in UNM and BEE. Soil pH ranged from 4.69 in SPR to 4.93 in UNM, indicating acidic soil. Soil pH, total C and N contents, and the C/N ratio did not differ significantly amongst the forests (Table 1).

**Table 1.** Mean and standard deviation (SD) of soil moisture, soil pH (H$_2$O), soil carbon content (C%), soil nitrogen content (N%), and the proportion of carbon and nitrogen (C/N) in three forest types: BEE—European beech *Fagus sylvatica* L.; SPR—Norway spruce *Picea abies* (L.) H. Karst.; UNM—unmanaged old-growth forest, in 2019 in Pol'ana Mts.

| | F(2,44) | p | | BEE | | | SPR | | | UNM | | |
|---|---|---|---|---|---|---|---|---|---|---|---|---|
| | | | | Mean | SD | | Mean | SD | | Mean | SD | |
| Soil moisture | 4.619 | 0.015 | * | 54.86 | 15.31 | ab | 42.48 | 7.54 | a | 58.65 | 20.15 | b |
| pH-H$_2$O | 2.011 | 0.147 | | 4.86 | 0.35 | | 4.69 | 0.29 | | 4.93 | 0.34 | |
| C% | 0.210 | 0.811 | | 9.10 | 2.48 | | 9.27 | 2.14 | | 9.71 | 3.23 | |
| N% | 0.435 | 0.650 | | 0.77 | 0.17 | | 0.76 | 0.13 | | 0.82 | 0.24 | |
| C/N | 0.634 | 0.536 | | 11.66 | 1.07 | | 12.07 | 1.27 | | 11.71 | 0.84 | |

Results from one-way ANOVA with factor forest type (F and *p*-value with significance at $p < 0.05$(*)). Significant differences ($p < 0.05$) in mean values between forest types are indicated by different lowercase letters within a row (the Fisher LSD post hoc test).

A total of 51 nematode genera were identified in all soil samples. The number of nematode genera was the highest in BEE (46), followed by UNM (38) and SPR (37). *Rhabditis* and *Filenchus* were dominant (>10%) and frequent (100%) in BEE. *Acrobeloides*, *Plectus*, *Rhabditis*, and *Aphelenchoides* were dominant and frequent in SPR, and *Acrobeloides*, *Rhabditis*, and *Filenchus* were dominant and frequent in UNM (Table 2).

**Table 2.** List of identified nematode genera, their dominance D%, and frequency of occurrence F% at three forests: BEE—European beech *Fagus sylvatica* L.; SPR—Norway spruce *Picea abies* (L.) H. Karst.; UNM—unmanaged old-growth forest, and at three age classes: 0–20-, 40–60-, and 100–120-year-old BEE and SPR forests. Nematode genera are classified by order according to the most recent and comprehensive system of free-living nematodes presented by Andrassy (2005, 2007, 2009). Nematode genera are evaluated by trophic groups (TG) using online program 'NINJA: An automated calculation system for nematode-based biological monitoring' (Sieriebriennikov et al., 2014): bacterivores—B; fungivores—F; herbivores—H; omnivores—O; predators—P.

| Nematode Genera | TG | Forest Stand | | | | | | Age (BEE + SPR) | | | | | |
| --- | --- | --- | --- | --- | --- | --- | --- | --- | --- | --- | --- | --- | --- |
| | | BEE | | SPR | | UNM | | 0–20 year | | 40–60 Year | | 100–120 Year | |
| | | D% | F% | D% | F% | D% | F% | D% | F% | D% | F% | D% | F% |
| Araeolaimida | | | | | | | | | | | | | |
| *Aulolaimus* | B | 0.2 | 13 | - | - | - | - | 0.1 | 10 | 0.1 | 10 | - | - |
| *Chronogaster* | B | - | - | 0.1 | 13 | - | - | 0.2 | 20 | - | - | - | - |
| *Plectus* | B | 6.6 | 100 | 16.3 | 100 | 6.6 | 100 | 15.6 | 100 | 6.9 | 100 | 12.2 | 100 |
| *Ceratoplectus* | B | 0.7 | 40 | - | - | - | - | 0.6 | 30 | 0.1 | 30 | - | - |
| *Wilsonema* | B | 1.7 | 80 | 0.7 | 47 | 2.0 | 67 | 1.7 | 70 | 0.9 | 70 | 0.5 | 50 |
| *Euteratocephalus* | B | 0.1 | 7 | - | - | - | - | 0.1 | 10 | - | - | - | - |
| Rhabditida | | | | | | | | | | | | | |
| *Teratocephalus* | B | 2.2 | 87 | 1.4 | 40 | 0.3 | 27 | 3.4 | 100 | 0.7 | 50 | 0.6 | 40 |
| *Cephalobus* | B | 2.3 | 73 | 2.8 | 80 | 3.0 | 67 | 1.5 | 90 | 1.4 | 50 | 5.6 | 90 |
| *Eucephalobus* | B | 0.4 | 20 | 0.6 | 47 | 2.0 | 80 | 1.0 | 60 | - | - | 0.4 | 40 |
| *Acrobeloides* | B | 8.3 | 100 | 21.3 | 100 | 16.7 | 73 | 14.2 | 100 | 9.9 | 100 | 23.8 | 100 |
| *Chiloplacus* | B | 0.4 | 53 | 0.2 | 20 | - | - | 0.3 | 40 | 0.3 | 40 | 0.2 | 30 |
| *Cervidellus* | B | 6.2 | 93 | 6.7 | 87 | 5.0 | 87 | 5.0 | 100 | 6.4 | 90 | 8.8 | 80 |
| *Acrobeles* | B | 0.5 | 20 | 0.1 | 7 | 0.1 | 7 | 0.5 | 30 | 0.1 | 10 | - | - |
| *Rhabditis* | B | 17.3 | 100 | 12.8 | 100 | 15.2 | 100 | 14.7 | 100 | 20.8 | 100 | 7.9 | 100 |
| *Diploscapter* | B | - | - | 0.6 | 40 | 1.8 | 33 | - | - | 0.3 | 30 | 0.8 | 30 |
| *Steinernema* | B | 0.7 | 27 | - | - | - | - | 0.7 | 30 | - | - | 0.1 | 10 |
| Aphelenchida | | | | | | | | | | | | | |
| *Aphelenchus* | F | 1.3 | 33 | 0.3 | 13 | 0.1 | 7 | 0.9 | 40 | 0.6 | 20 | 0.5 | 10 |
| *Aphelenchoides* | F | 3.3 | 100 | 10.1 | 100 | 4.1 | 67 | 6.4 | 100 | 7.7 | 100 | 7.3 | 100 |
| Tylenchida | | | | | | | | | | | | | |
| *Aglenchus* | H | 1.1 | 40 | 0.1 | 7 | 0.9 | 27 | - | - | 0.3 | 30 | 1.5 | 40 |
| *Coslenchus* | H | 0.2 | 20 | 0.1 | 7 | - | - | - | - | 0.2 | 10 | 0.3 | 30 |
| *Filenchus* | F | 11.2 | 100 | 5.8 | 100 | 10.1 | 100 | 6.0 | 100 | 10.1 | 100 | 9.4 | 100 |
| *Tylenchus* | H | 0.1 | 7 | - | - | - | - | 0.2 | 10 | - | - | - | - |
| *Malenchus* | H | 6.2 | 73 | 1.9 | 87 | 6.0 | 93 | 2.8 | 80 | 6.8 | 80 | 2.0 | 80 |
| *Tylenchorhynchus* | H | 0.4 | 20 | 0.4 | 13 | 0.2 | 13 | 0.9 | 40 | 0.1 | 10 | - | - |
| *Pratylenchus* | H | 0.3 | 27 | 0.2 | 27 | 0.2 | 7 | 0.3 | 20 | 0.1 | 10 | 0.5 | 50 |
| *Helicotylenchus* | H | 0.5 | 33 | 0.3 | 60 | 0.7 | 33 | 0.6 | 50 | 0.2 | 60 | 0.4 | 40 |
| *Rotylenchus* | H | 0.2 | 27 | 0.2 | 20 | 0.3 | 7 | 0.2 | 20 | 0.1 | 20 | 0.2 | 30 |
| *Heterodera* | H | 0.1 | 7 | - | - | 0.3 | 27 | - | - | - | - | 0.1 | 10 |
| *Paratylenchus* | H | 5.7 | 100 | 1.6 | 73 | 4.1 | 67 | 1.6 | 80 | 6.2 | 80 | 2.7 | 100 |
| *Mesocriconema* | H | 0.1 | 20 | - | - | - | - | - | - | 0.1 | 30 | - | - |
| Enoplida | | | | | | | | | | | | | |
| *Prismatolaimus* | B | 4.5 | 87 | 2.4 | 73 | 1.3 | 67 | 4.5 | 80 | 2.6 | 90 | 2.3 | 70 |
| *Tripyla* | P | 1.2 | 87 | 0.9 | 73 | 1.1 | 60 | 1.1 | 90 | 1.1 | 80 | 0.8 | 70 |
| Alaimida | | | | | | | | | | | | | |
| *Alaimus* | B | 1.4 | 93 | 1.3 | 93 | 2.0 | 87 | 1.2 | 90 | 1.8 | 100 | 0.9 | 90 |
| *Amphidelus* | B | 0 | 7 | - | - | 0.1 | 7 | - | - | 0.1 | 10 | - | - |
| Diphtherophorida | | | | | | | | | | | | | |
| *Diphtherophora* | F | 0.6 | 60 | 0.2 | 20 | 0.3 | 27 | 0.4 | 30 | 0.4 | 60 | 0.1 | 30 |
| *Trichodorus* | H | 0.1 | 13 | 0.6 | 47 | 0.9 | 33 | 0.5 | 50 | 0.6 | 40 | - | - |
| Mononchida | | | | | | | | | | | | | |
| *Clarkus* | P | - | - | - | - | 1.6 | 60 | - | - | - | - | - | - |
| *Prionchulus* | P | 0.1 | 7 | - | - | 0.2 | 13 | 0.1 | 10 | - | - | - | - |
| *Mylonchulus* | P | 1.5 | 73 | 1.2 | 73 | 0.7 | 27 | 1.9 | 90 | 1.0 | 70 | 0.9 | 60 |
| *Anatonchus* | P | 0.5 | 80 | 0.5 | 53 | 1.2 | 67 | 0.2 | 60 | 0.7 | 80 | 0.7 | 60 |
| Dorylaimida | | | | | | | | | | | | | |
| *Dorylaimus* | O | 0.1 | 7 | - | - | - | - | 0.1 | 10 | - | - | - | - |
| *Mesodorylaimus* | O | 1.5 | 73 | 0.7 | 53 | 1.2 | 53 | 1.5 | 70 | 0.9 | 90 | 0.5 | 30 |
| *Discolaimus* | P | - | - | - | - | 0.1 | 7 | - | - | - | - | - | - |
| *Crassolabium* | O | 0.1 | 7 | - | - | 0.3 | 13 | 0.1 | 10 | - | - | - | - |
| *Eudorylaimus* | O | 8.0 | 93 | 6.8 | 100 | 7.4 | 100 | 7.0 | 90 | 8.8 | 100 | 6.1 | 100 |
| *Aporcelaimellus* | O | 0.5 | 47 | 0.4 | 33 | 0.3 | 27 | 0.3 | 40 | 0.5 | 40 | 0.6 | 40 |
| *Enchodelus* | P | - | - | 0.2 | 13 | 0.4 | 13 | 0.3 | 20 | - | - | - | - |
| *Xiphinema* | H | 0.1 | 13 | 0.1 | 7 | - | - | 0.1 | 10 | 0 | 10 | 0.1 | 10 |
| *Axonchium* | H | 0.2 | 33 | 0.1 | 13 | - | - | 0.2 | 40 | 0 | 10 | 0.1 | 20 |
| *Tylencholaimus* | F | 1.6 | 87 | 0.5 | 40 | 1.3 | 53 | 1.1 | 60 | 0.6 | 70 | 1.1 | 60 |
| *Doryllium* | F | 0.4 | 13 | - | - | - | - | 0.1 | 10 | 0.4 | 10 | - | - |
| Numbers of genera | | 46 | | 37 | | 38 | | 43 | | 39 | | 34 | |

Total nematode abundance (311 individuals/100 g) and biomass (0.61 mg) were lower in UNM than BEE and SPR (both $p < 0.001$). H′gen was lower in SPR than UNM and BEE, but not significantly (Table 3).

**Table 3.** The one-way ANOVA results (F and *p*-values with significance level) among three forest types: BEE—European beech *Fagus sylvatica* L.; SPR—Norway spruce *Picea abies* (L.) H. Karst.; UNM—unmanaged old-growth forest, in 2019 in Poľana Mts. Means and standard deviation (SD) of nematode abundance, genera diversity index, H'gen, abundance of nematodes in trophic groups—ecological and functional indices. Soil sampling in 2019 in Pol'ana Mts., Slovak Republic.

| | F(2,44) | *p* | | BEE Mean | BEE SD | | SPR Mean | SPR SD | | UNM Mean | UNM SD | |
|---|---|---|---|---|---|---|---|---|---|---|---|---|
| Abundance | 21.502 | 0 | *** | 631.07 | 176.79 | a | 778.33 | 239.53 | a | 311 | 207.14 | b |
| H'gen | 2.529 | 0.092 | | 2.40 | 0.46 | | 2.24 | 0.25 | | 2.40 | 0.22 | |
| Total Biomass, mg | 8.984 | 0.001 | *** | 1.51 | 1.19 | a | 1.25 | 0.85 | a | 0.61 | 0.58 | b |
| Bacterivores% | 4.675 | 0.015 | * | 52.48 | 15.21 | a | 65.49 | 12.04 | b | 49.53 | 17.76 | a |
| Fungivores% | 0.179 | 0.837 | | 18.67 | 5.97 | | 17.17 | 6.31 | | 17.72 | 9.17 | |
| Herbivores% | 6.802 | 0.003 | ** | 15.48 | 13.99 | a | 6.70 | 6.4 | b | 17.93 | 10.79 | a |
| Omnivores% | 0.771 | 0.469 | | 10.20 | 4.24 | | 7.93 | 5.02 | | 9.81 | 6.57 | |
| Predators% | 1.06 | 0.356 | | 3.17 | 1.32 | | 2.69 | 1.97 | | 5.01 | 4.76 | |
| Maturity Index | 1.257 | 0.295 | | 2.27 | 0.27 | | 2.18 | 0.22 | | 2.33 | 0.34 | |
| Plant Parasitic Index | 0.759 | 0.474 | | 2.30 | 0.33 | | 2.43 | 0.37 | | 2.33 | 0.47 | |
| Enrichment Index | 1.903 | 0.162 | | 61.05 | 17.74 | | 49.5 | 19.06 | | 57.81 | 15.87 | |
| Structure Index | 4.304 | 0.02 | * | 65.65 | 10.95 | b | 46.39 | 19.44 | a | 58.44 | 21.6 | ab |
| Basal Index | 3.381 | 0.044 | * | 22.11 | 9.44 | a | 35.93 | 17.27 | b | 26.38 | 14.46 | ab |
| Channel Index | 0.16 | 0.852 | | 28.09 | 19.41 | | 29.73 | 15.81 | | 27.86 | 19.8 | |

Significance level * $p = 0.05$; ** $p = 0.01$; *** $p = 0.001$. Significant differences ($p < 0.05$) in mean values between forest stands are indicated by different lowercase letters within a row (the Fisher LSD post hoc test).

The evaluation of the proportions of nematodes in the trophic groups identified significant differences between bacterivores and herbivores. Bacterivore abundance was significantly higher in SPR than in BEE and UNM ($p < 0.05$), and herbivore abundance was significantly higher in UNM and BEE than in SPR ($p < 0.01$). The number of predator genera and the proportion of predators were highest in UNM (Table 2), but not significantly (Table 3).

The ecological indices (MI and PPI) and functional indices (EI, SI, BI, and CI) varied amongst the samples but did not differ significantly between the studied forests. The SI was lower, and the BI was higher in SPR than in BEE (both $p < 0.05$), and both indices were similar to those in UNM (Table 3).

A plot of the EI and SI, and a graphical display of the functional metabolic footprints of the conditions of the soil food web (Figure 1) indicated that most samples from BEE and UNM were in quadrat B, indicating a maturing N-rich environment with a low C/N ratio and multiple pathways of decomposition. The samples from SPR were mostly in quadrat D, indicating a degraded environment with a fungal pathway of decomposition.

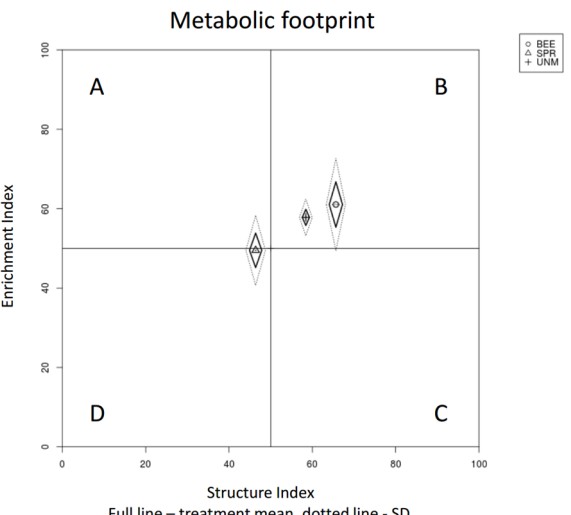

**Figure 1.** Functional metabolic footprint of nematodes in three forest types: BEE—European beech *Fagus sylvatica* L.; SPR—Norway spruce *Picea abies* (L.) H. Karst.; UNM—unmanaged old-growth forest, in 2019 in Poľana Mts.

### 3.2. Effect of Forest Age Class

The soil properties only differed between the three age classes (for both BEE and SPR) for soil acidity, which was higher in the 100–120-year-old forest than in the younger forests (both $p < 0.05$). The other soil parameters did not differ significantly between the age classes (Table 4).

**Table 4.** Mean and standard deviation (SD) of soil moisture, soil pH ($H_2O$), soil carbon content (C%), soil nitrogen content ($n$%), and the proportion of carbon and nitrogen (C/N) at two stands: BEE—European beech *Fagus sylvatica* L.; SPR—Norway spruce *Picea abies* (L.) H. Karst., and in age classes: 0–20-, 40–60-, and 100–120-year-old forests, in 2019 in Poľana Mts.

| | Stand | | Age | | BEE | | | SPR | | | 0–20 Year | | | 40–60 Year | | | 100–120 Year | | |
|---|---|---|---|---|---|---|---|---|---|---|---|---|---|---|---|---|---|---|---|
| | F(1,26) | | F(2,26) | | Mean | SD | | Mean | SD | | Mean | SD | | Mean | SD | | Mean | SD | |
| Soil moisture | 8.052 | ** | 1.272 | | 54.86 | 15.31 | a | 42.48 | 7.54 | b | 46.28 | 8.20 | | 46.14 | 13.16 | | 53.59 | 17.32 | |
| pH-$H_2O$ | 2.779 | | 6.721 | ** | 4.86 | 0.35 | | 4.69 | 0.29 | | 4.96 | 0.28 | a | 4.84 | 0.32 | a | 4.53 | 0.23 | b |
| Total C | 0.04 | | 0.882 | | 9.10 | 2.48 | | 9.27 | 2.14 | | 8.64 | 1.60 | | 9.96 | 2.78 | | 8.96 | 2.32 | |
| Total N | 0.045 | | 1.696 | | 0.77 | 0.17 | | 0.76 | 0.13 | | 0.70 | 0.10 | | 0.82 | 0.19 | | 0.77 | 0.13 | |
| C/N | 0.903 | | 1.167 | | 11.66 | 1.07 | | 12.07 | 1.27 | | 12.22 | 1.12 | | 11.93 | 0.97 | | 11.43 | 1.38 | |

Results from main effect ANOVA (F and *p*-value with significance at $p < 0.01$(**) for both factors: stand and age). Significant differences ($p < 0.05$) in mean values between age classes are indicated by different lowercase letters within a row (the Fisher LSD post hoc test). Significant differences between stands are marked.

The number of nematode genera was higher in the 0–20-year-old forest (43) than the 40–60-year-old year forest (39) and the 100–120-year-old forest (34). The dominant genera with frequencies of occurrence of 100% were *Plectus* (16%), *Rhabditis* (15%), and *Acrobeloides* (14%) in the 0–20-year-old forest, *Rhabditis* (21%) and *Filenchus* (10%) in the 40–60-year-old forest, and *Acrobeloides* (24%) and *Plectus* (12%) in the 100–120-year-old forest (Table 2).

The number of nematode genera and total nematode abundance and biomass were highest in the 0–20-year-old forests in both BEE and SPR (both $p < 0.01$), and H'gen was also highest in these forests, but not significantly. The SI was higher, and the BI was lower in the 0–20- and 0–40-year-old forests than in the 100–120-year-old forest in both BEE and SPR (all $p < 0.01$). The MI, PPI, and CI varied but were not correlated with stand age. Bacterivores were the most abundant trophic group in all age classes, followed by fungivores and herbivores. The abundances of the trophic groups did not differ significantly amongst the age classes (Table 5).

**Table 5.** Main effect ANOVA results for factors 'stand' (BEE—European beech *Fagus sylvatica* L.; SPR—Norway spruce *Picea abies* (L.) H. Karst.) and 'age' (0–20-, 40–60-, and 100–120-year-old forests). Mean and standard deviation (SD) of nematode abundance, genera diversity index, H'gen, abundance of nematodes in trophic groups—ecological and functional indices. Soil sampling in 2019 in Poľana Mts., Slovak Republic.

| | Stand | | Age | | BEE | | | SPR | | | 0–20 Year | | | 40–60 Year | | | 100–120 Year | | |
|---|---|---|---|---|---|---|---|---|---|---|---|---|---|---|---|---|---|---|---|
| | F(1,26) | | F(2,26) | | Mean | SD | | Mean | SD | | Mean | SD | | Mean | SD | | Mean | SD | |
| Abundance | 4.781 | * | 6.469 | ** | 631.07 | 176.79 | a | 778.33 | 239.53 | b | 863.2 | 230.13 | a | 677.0 | 187.42 | b | 573.9 | 141.69 | b |
| H'gen | 6.307 | * | 2.786 | | 2.40 | 0.46 | | 2.24 | 0.25 | | 2.48 | 0.30 | | 2.22 | 0.49 | | 2.27 | 0.30 | |
| Total Biomass, mg | 0.346 | | 7.347 | ** | 1.51 | 1.19 | | 1.25 | 0.85 | | 1.94 | 1.27 | a | 1.49 | 0.94 | a | 0.72 | 0.19 | b |
| Bacterivores% | 7.505 | * | 2.674 | | 52.48 | 15.21 | a | 65.49 | 12.04 | b | 63.81 | 9.60 | | 51.23 | 17.34 | | 61.91 | 15.2 | |
| Fungivores% | 0.515 | | 1.242 | | 18.67 | 5.97 | | 17.17 | 6.31 | | 15.47 | 4.34 | | 19.05 | 6.53 | | 19.25 | 6.92 | |
| Herbivores% | 5.365 | * | 0.688 | | 15.48 | 13.99 | a | 6.70 | 6.40 | b | 8.12 | 6.04 | | 16.6 | 16.76 | | 8.55 | 8.01 | |
| Omnivores% | 2.949 | | 1.292 | | 10.20 | 4.24 | | 7.93 | 5.02 | | 9.18 | 3.98 | | 10.3 | 5.31 | | 7.72 | 4.86 | |
| Predators% | 1.581 | | 1.141 | | 3.17 | 1.32 | | 2.69 | 1.97 | | 3.43 | 1.33 | | 2.81 | 1.84 | | 2.54 | 1.81 | |
| Maturity Index | 1.472 | | 0.275 | | 2.27 | 0.27 | | 2.18 | 0.22 | | 2.27 | 0.25 | | 2.20 | 0.32 | | 2.21 | 0.18 | |
| Plant Parasitic Index | 1.213 | | 0.922 | | 2.30 | 0.33 | | 2.43 | 0.37 | | 2.49 | 0.41 | | 2.35 | 0.37 | | 2.27 | 0.26 | |
| Enrichment Index | 4.39 | * | 6.628 | ** | 61.05 | 17.74 | a | 49.5 | 19.06 | b | 54.27 | 18.16 | ab | 68.19 | 14.34 | a | 43.37 | 16.92 | b |
| Structure Index | 14.76 | *** | 5.985 | ** | 65.65 | 10.95 | a | 46.39 | 19.44 | b | 59.53 | 13.14 | a | 64.34 | 17.76 | a | 44.18 | 18.68 | b |
| Basal Index | 9.556 | ** | 6.968 | ** | 22.11 | 9.44 | a | 35.93 | 17.27 | b | 26.65 | 11.74 | a | 20.65 | 11.8 | a | 39.76 | 16.49 | b |
| Channel Index | 0.261 | | 2.689 | | 28.09 | 19.41 | | 29.73 | 15.81 | | 25.95 | 17.21 | | 22.25 | 11.05 | | 38.53 | 19.92 | |

F and *p*-value with significance level (* $p = 0.05$; ** $p = 0.01$; *** $p = 0.001$) for both factors: 'stand' and 'age'. Significant differences between stands ($n = 15$, $p < 0.05$). Significant differences ($p < 0.05$) in mean values ($n = 5$) between age classes are indicated by different lowercase letters within a row (the Fisher LSD post hoc test).

When plotting the EI and SI, and the functional metabolic footprint of nematodes, most soil samples were from the 0–20-year-old and 40–60-year-old forests, suggesting that the food webs were enriched and structured, with both bacterial and fungal pathways of decomposition and maturing food webs—quadrat B. The 100–120-year-old forests, however, were characterized as degraded environments with fungal pathways of decomposition—quadrat D (Figure 2).

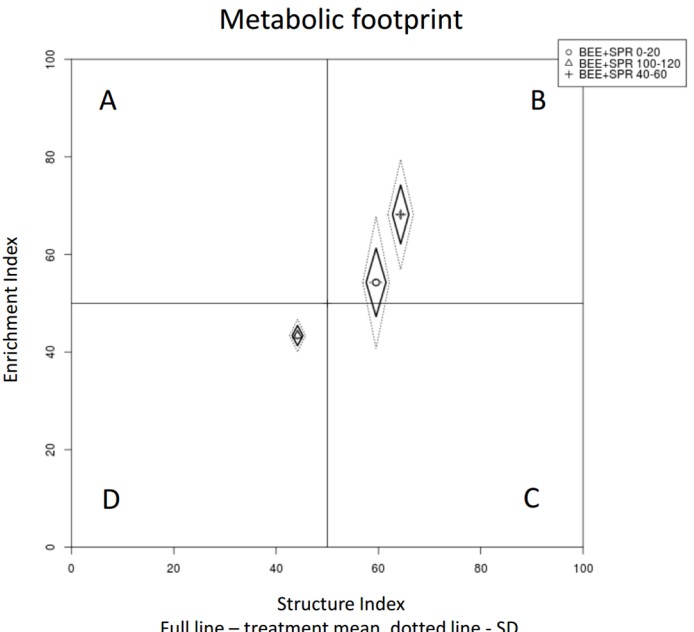

**Figure 2.** Functional metabolic footprint of nematodes at three different age classes: 0–20-, 40–60-, and 100–120-year-old forests, in 2019 in Pol'ana Mts.

### 3.3. Mixed Effect of Forest Management × Age

The mixed effect of forest management × age was significant only for H'gen, PPI, and the proportion of omnivores. A one-way ANOVA was performed separately for each type of forest.

Omnivore proportions differed amongst the SPR age classes. The proportions were higher in the 40–60-year-old forest (12.68%) than the 0–20-year-old forest (6.76%) and the 100–120-year-old forest (5.61%) (both $p < 0.05$).

In BEE, H'gen was higher in the 0–20-year-old forest (2.68) than the 40–60-year-old forest (2.03) ($p < 0.05$), and PPI was higher in the 0–20-year-old forest (2.65) than the 40–60-year-old forest (2.12) and the 100–120-year-old forest (2.15) (both $p < 0.05$).

## 4. Discussion

### 4.1. Effect of Forest Management

Depending on the silvicultural system, forest management may strongly affect the soil quality and functioning that generally correlate with the successional stage of forest vegetation. Clearcutting and plantation forestry alter the forest structure, composition, and diversity, and thus the understory vegetation changes depending on the age and density of the forest stand. Canopy reduction and removal are usually reflected in the changes in soil temperature and moisture content [46–49], the input of organic matter, and the C and N contents [50,51]. Soil nematodes need soil moisture for movement, feeding, and reproduction [52]. The soil moisture content in our study was the highest in UNM, but the total nematode abundance and biomass were only half those of the managed forests. Some previous works [53,54] reported a higher nematode abundance and diversity in old-growth forests, but our results did not support this observation. The harvesting of wood in a managed forest removes vegetation that could be added to the soil to feed microorganisms

(bacteria, fungi, protozoa, and nematodes). When plants die, soil microbes break down plant C compounds and use them for metabolism and growth, respiring some of the C back to the atmosphere. Microbial decomposition releases carbon dioxide ($CO_2$), meaning the soil can store more C when it is protected from microbial activity. Dead wood is retained in UNM and decomposes in the soil, and soil microorganisms, including nematodes, break down its C compounds and uses them for metabolism and growth. Soil nematodes respond to elevated levels of $CO_2$, and an increase in $CO_2$ concentrations can negatively affect the nematode abundance in forest soil [55]. The lower total nematode abundance and biomass in UNM may have also been due to the balanced nematode communities in the stable environment, without substantial dominance by any one trophic group. The abundance and number of predator genera were higher in UNM than the managed forests. An increase in trophic levels by slow-growing nematode species such as omnivores and predators has been linked to soil stability and maturity [21].

The number of nematode genera was lower, and nematode abundance was higher in SPR than in BEE or UNM. The higher total nematode abundance in SPR was mostly due to the high abundance of bacterivorous nematodes. These results suggest higher microbial activity and a higher supply of N in the environment [23,41]. The fixation of atmospheric $N_2$ leads to higher levels of N in and around plants [56], which contribute to the rapid growth of bacteria that could account for the high abundances of bacterivorous nematodes. The abundance of bacterivorous nematodes usually increases in nutrient-rich but disturbed forest soil, e.g., due to windstorms, wildfires, or clearcutting [57,58], although other authors reported a higher abundance of bacterivores in healthy than disturbed forest soil [59].

An increase in the number of nematode taxa is usually associated with an increase in the number of plant species [60], and in our study, the highest number of nematode genera was observed in the deciduous forests. A higher proportion of herbivores is usually associated with forest disturbances [59,61], and in our study, we confirmed a higher proportion of herbivores in BEE than in SPR. The richness of herbivorous nematodes is usually the highest at sites with extensive shrub cover, a high sand content, and a low pH, but also in old forests [62], and the abundance of herbivorous nematodes also depends on the type of herbaceous undergrowth, whose roots they can feed on. Herbivorous nematodes usual prefer to attack grass roots over the more solid roots of conifers [63,64].

Integrating trophic groups and life strategies into functional guilds has allowed the definition of several indices that describe the structure, function, and condition of food webs in disturbed or stressed environments [40]. Neither the MI nor the PPI in our study was affected by forest management or age. Previous studies reported a lower MI in clearcut [57] and managed forests than in natural forests [65], or during recovery after disturbances [66]. Other studies, however, have reported that the MI in forest soil was a better indicator of site production and the harshness of microclimatic conditions than the degree of disturbance [58,59,67]. The SI represents the relative contribution of nematodes with high cp values (3–5) and indicates the state of food webs affected by stress or disturbance. Soil disturbances subsequently lead to lower SIs. A high BI, though, represents a high abundance of taxa tolerant to stress with a moderately long lifecycle [41]. In our study, the SI was lower, and the BI was higher in the coniferous SPR forests than the deciduous BEE forests.

### 4.2. Effect of Forest Age Class

Upland stands form closed canopies, accumulate litter, and lose coarse woody debris over time, all changes that are expected in a growing forest. Various organic acids are produced during the decomposition of dead wood, which can lower the soil pH. The soil pH influences nutrient uptake and tree growth and tends to be lower in old forests than comparable young managed or naturally seeded sites [62,65]. The soil pH only had minor changes in our study and ranged from 4.5 to 4.9.

The nematode abundance and diversity in managed forests are correlated with the successional stage of the forest vegetation [24,68–70]. Older stands usually had more

nematode species than younger stands due to the late appearance of rare and localised species [71], but a management strategy that increased the growth of the understory and the herbal layer could also stimulate nematode populations in the soil [54]. The nematode abundance and richness accompanying aboveground successions at 99 commercial forest sites, however, had no significant trends [62]. Some studies have concluded that high species richness in soils leads to functional redundancy where multiple species perform similar roles in ecosystems [72–74], but [75] cited several examples where species played specific roles in ecosystem functioning. The effects of diversity may be strongest at the species-poor end of diversity gradients where only a few species are present [76,77]. The number of nematode genera, total nematode abundance and biomass, and diversity in our study were the highest in the 0–20-year-old forests in both SPR and BEE, and all these parameters decreased as the forests grew and aged. The EI and SI in both SPR and BEE tended to be higher (>50%) in the younger (0–20 and 40–60 years old) than the 100–120-year-old-forests, indicating a nutrient-rich soil ecosystem where organic matter is decomposed by bacteria [41], omnivores and predators are less common, and food webs are degraded.

## 5. Conclusions

Our study identified important relationships amongst types of forest management, age classes, and soil nematode communities. The trees affected the composition of the nematode communities more than the community indices by affecting the nematode genera and a range of trophic groups. The species-rich deciduous forests supported by extensive root systems had a higher diversity of nematode taxa and a higher abundance of herbivores than the species-poor coniferous forests. Nematode abundance and biomass were lower in the old managed and the unmanaged forests, perhaps associated with the more extensive root systems or the sequestration of soil C. These findings suggest strong bottom-up effects of belowground tree inputs and indicate that some components of the nematode community may be differentially affected by the resource quantity and quality. Our data are currently limited to managed and natural forest sites in one biogeoclimatic zone. Additional sampling over a broader range of sites and forest types would greatly improve confidence in the use of nematode communities as indicators of soil health in forests.

**Author Contributions:** A.Č. and M.R. wrote the paper, D.M. analysed the data, and E.G. conceived, designed, and performed the experiments. All authors have read and agreed to the published version of the manuscript.

**Funding:** This study was supported by projects of the Slovak scientific agency VEGA, project No. 2/0018/20 (0.4), and the Slovak research and development agency, project numbers APVV 15-0176 (0.3) and APVV19-0142 (0.3). The authors would like to thank Marián Homolák for the organisation of and help with soil sampling.

**Institutional Review Board Statement:** Not applicable.

**Informed Consent Statement:** Not applicable.

**Data Availability Statement:** The data in this study are available in presented Tables.

**Conflicts of Interest:** The authors declare that they have no conflict of interest.

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
