# Peer review of "Soil Nematode Communities in Managed and Natural Temperate Forest"

_diversity, doi:10.3390/d13070327_

Round 1
Reviewer 1 Report
The paper by Čerevková et al. on “
Soil nematode communities in managed and natural temperate forest ” is a very interesting paper dealing with the influence of tree species in nematode communities. The manuscript is presenting new and significant information with interest for nematologists, ecologists, and foresters. The paper is well-organized and has been carried out following all the international standards and deserves publication in Diversity.
I do not have major corrections or suggestions, since the manuscript is very well curated in the experimental procedures and the English style.
See below my minor suggestions:
L74, I suspect that authors try to refers to fungivorous and not to frugivorous......
I suggest to authors compare their results with those of a similar recent study in a cultivated ecosystem, such as cultivated olive in southern Europe: viz. Agriculture, Ecosystems & Environment, Volume 287, 1 January 2020, 106688
References section: from Line 460 onwards remove the double reference numeration....
Author Response
Dear Reviewer
thanks for the comments and corrections that helped improve the quality of the manuscript. Your correction (Line 74 and References) was added. The Paper recommended for the discussion is very interesting and I will use it in our next paper.
Reviewer 2 Report
In the revised article entitled ‘Soil nematode communities in managed and natural 2 temperate forest’, submitted to DIVERSITY by Andrea Čerevková and Co-authors have presented results of interesting study on nematodes communities in various forests.
Generally, I find the results valuable as they include 1) three type of forests (beech, spruce and mixed), 2) soil environmental factors (soil parameters) 3) two forests management systems (managed and unmanaged) 4) forests in three age classes in a single study. In my opinion the introduction is complete and give broader context, goals of the study are specified, and results are clearly presented. I have mainly editorial comment which are included in the attached PDF file.

Author Response
Dear Reviewer,
thank you for all your comments that all were used in the corrected manuscript.
English grammar and spelling in the submitted manuscript was edited by Dr. William Blackhall, Editor, Global Biological Editing https://www.globalbiologicalediting.com/index.html
